# The Long-Term Effect of COVID-19 Infection on Body Composition

**DOI:** 10.3390/nu16091364

**Published:** 2024-04-30

**Authors:** Ornina Atieh, Jared C. Durieux, Jhony Baissary, Christian Mouchati, Danielle Labbato, Alicia Thomas, Alexander Merheb, Kate Ailstock, Nicholas Funderburg, Grace A. McComsey

**Affiliations:** 1School of Medicine, Case Western Reserve University, Cleveland, OH 44106, USA; ornina.atieh@case.edu (O.A.); jxb1120@case.edu (J.B.); or alicia.thomas@uhhospitals.org (A.T.); 2Center for Clinical Research, University Hospitals Cleveland Medical Center, Cleveland, OH 44106, USA; jared.durieux@uhhospitals.org (J.C.D.); danielle.labbato@uhhospitals.org (D.L.); 3Faculty of Arts and Sciences, Harvard College, Cambridge, MA 02138, USA; amerheb@college.harvard.edu; 4School of Health and Rehabilitation Sciences, Ohio State University, Columbus, OH 43210, USA; kate.ailstock@osumc.edu (K.A.); nicholas.funderburg@osumc.edu (N.F.)

**Keywords:** long-COVID, PASC, body composition, COVID-19 pandemic

## Abstract

Background: The effect of COVID-19 infection versus the indirect effect of the pandemic on body composition remains unclear. This study investigates the long-term changes in body composition in COVID-19 survivors compared to a contemporary control group. Method: This is a prospective study involving adults who underwent a pre-pandemic whole-body DXA scan (DXA#1) between 2017 and 2019. Participants were asked to return for a repeat whole-body DXA scan (DXA#2) after the pandemic. Detailed data were collected including their medical and COVID-19 history. Inflammation markers and fasting lipids were measured. For those participants who experienced a COVID-19 infection between the two DXAs, DXA#2 was acquired at least one year after COVID-19 infection. Results: Overall, 160 adults were enrolled; 32.5% females, 51.8% non-white, with mean age of 43.2 years. Half (*n* = 80) of the participants experienced a COVID-19 infection between their two DXA scans (COVID-19+ group), and the other half had never had COVID-19. COVID-19-negative participants displayed an increase in annualized trunk fat (g) [922.5 vs. 159.7; *p* = 0.01], total fat (g) [1564.3 vs. 199.9; *p* = 0.2], and LBM (g) [974.9 vs. −64.5; *p* = 0.0002] when compared to the COVID-19+ group. However, among the COVID-19+ group, no differences were seen in annualized trunk fat, total fat mass, or LBM between those with PASC and without (*p* > 0.05). Conclusion: During the pandemic, both the COVID-19 survivors and the COVID-19-negative group exhibited increases in weight, total fat, and trunk fat, likely associated with pandemic-linked lifestyle modifications. However, only COVID-19 survivors displayed a decline in lean body mass over the same period, regardless of PASC symptoms.

## 1. Introduction

The coronavirus disease-2019 (COVID-19) pandemic, caused by severe acute respiratory syndrome coronavirus-2 (SARS-CoV-2), has had a massive influence on health, including millions of individuals globally who suffer from long-COVID, also known as post-acute sequelae of SARS-CoV-2 (PASC) [1]. In addition to fatigue, brain fog, and the loss of sense of taste or smell, long-COVID also encompasses any newly onset condition apparently triggered by COVID-19 infection, such as new onset diabetes or autoimmune disease [2,3,4]. The multifaceted direct impact of COVID-19 on individuals, combined with the indirect consequences of lockdown measures and reduced physical activity, has raised concerns about alterations in body composition factors such as weight, fat, bone mineral density (BMD), and lean body mass (LBM) [5,6,7,8].

Studies conducted during and after the pandemic have focused on exploring the relationship between the severity of COVID-19 infection, lockdown, and body composition. Some of these studies have looked at the impact of body composition on COVID-19 outcomes and severity [9,10], while others have assessed the risks associated with the pandemic in relation to body composition outcomes after recovery. The first meta-analysis conducted after the initial lockdown in 2020 found a significant increase in body weight among individuals aged 16 years and older [5]. Another study reported a worsening in BMD, fat mass, and hand grip strength in women with unhealthy lifestyles after 13 weeks of lockdown [6]. 

While confinement, an unhealthy diet, increased alcohol intake, and reduced physical activity can contribute to a deterioration in overall health and body composition [11] and can induce sarcopenia [12,13], SARS-CoV-2 may influence these parameters as well. The mechanism of action of this virus on human cells was identified by multiple studies [14,15]. This lineage B Beta-coronavirus can enter human cells expressing angiotensin-converting enzyme 2 (ACE2) receptors including bone and muscle cells [14,16,17]. ACE2 receptors regulate inflammation by converting angiotensin II (inflammatory) to angiotensin 1–7 (anti–inflammatory) [17]. Upon infection, SARS-CoV-2 infiltrates specific host cells via spike protein–ACE2 receptor binding, leading to a decrease in ACE2 expression with subsequent elevated angiotensin-II levels. This triggers extensive inflammation in the affected organs [17,18]. Additionally, the immune response to SARS-CoV-2 infection may contribute to musculoskeletal damage via the excessive production of pro-inflammatory cytokines such as IL-6, IL-17, IL-18, TNF-α, and MCP1 [17,19]. The literature extensively documents the muscular consequences of the initial acute COVID-19 infection, including myalgia, muscle weakness, and loss, as well as reduced physical performance post-COVID-19 infection [20,21]. 

Few studies have evaluated changes in body composition in individuals with long-COVID. Sampalo et al. described a high prevalence of sarcopenia three months post-COVID-19 infection and an increase in weight and fat mass after 1 year [7]. In a cross-sectional study conducted by Velez R comparing individuals with long-COVID to a control group, those with long-COVID exhibited more total fat and visceral adiposity, along with lower lean muscle levels [8]. 

To our knowledge, no one has studied the long-term effects of COVID–19 infection on body composition, particularly in comparison to pre-infection baseline data and an uninfected control group from the same pandemic period. This study aims to investigate, for the first time, the long-term changes in LBM, total BMD, and total and trunk fat in COVID-19 survivors compared to a control group of never infected participants using pre-pandemic and post-pandemic DXA scans accounting for COVID-19 infection status.

## 2. Methods

### 2.1. Study Design 

Participants with both available pre-pandemic data, including a whole-body DXA scan, obtained between 2017 and 2019, and post-pandemic DXA data were selected for this study. A total of 160 participants met these criteria, consisting of 80 COVID-19 survivors and 80 participants who had never experienced a SARS-CoV-2 infection. The whole-body DXA scan available from the pre-pandemic period is referred to as DXA#1, while the post-pandemic DXA scan is referred to as DXA#2. No power calculation was needed given the design of this study. Comprehensive demographics, clinical data including medical history, and inflammation markers were obtained at DXA#1 and DXA#2. Fasting lipids and creatine kinase (CK) levels were assessed in real-time in a CLIA-certified laboratory at University Hospitals Cleveland Medical Center (UHCMC), Cleveland, Ohio.

### 2.2. Recruiting and Study Population

All participants who had previously been enrolled in studies conducted at the Metabolic Research Center of UHCMC were approached if they had expressed willingness to be contacted for future research projects, were at least 18 years of age, and underwent a whole-body DXA scan between 2017 and 2019.

In addition, to be included, participants had to either (1) have no history of COVID-19 or suggestive symptoms/illness since DXA#1, with negative antibody tests (negative nucleocapsid antibody and negative spike antibody if unimmunized) performed at the post-pandemic evaluation (COVID-19-negative group) or (2) have at least one documented COVID-19 infection (confirmed by a positive PCR or antigen test, or documented in the electronic medical records (EMR)) since DXA#1 and the COVID-19 infection dated back at least 1 year prior to the post-pandemic visit. We excluded women who were pregnant or lactating and those with cognitive impairment unable to consent. This study was approved by the Institutional Review Board of the Institutional Review Board of University Hospitals Cleveland Medical Center, Cleveland, OH, USA (approval code STUDY20221484; approval date: 24 October 2023) and a signed consent form was obtained from each participant before any study activity. 

### 2.3. Study Measurements

#### 2.3.1. Baseline Characteristics

The same questionnaires were used for all participants to obtain the following information: demographic characteristics (age, race/ethnicity, and sex), smoking and alcohol consumption, family history (cardiovascular disease, diabetes, hypertension, strokes, or hip fracture), and the level of physical activity (in minutes/week). A detailed review of their medical history was obtained from participants and their EMRs, including co-morbidities, clinical diagnoses, COVID-19 infection(s) dates, tests performed, and course of illness. Weight and height were measured using standardized protocol [22]. 

The group of individuals who had at least 1 COVID-19 infection (COVID-19+ group) was divided based on whether they displayed symptoms of PASC or not (referred to, respectively, as COVID-19+ PASC+ or COVID-19+ No PASC). PASC was considered as having ≥ 2 persistent symptoms that started after COVID-19 infection and lasted for at least 3 months, and the symptoms captured in the questionnaires filled out by patients included fatigue, loss of energy, shortness of breath, cough, change or loss of smell or taste, feeling pain in any part of the body, trouble thinking or concentrating, feeling sad or anxious, sweat chills and fever, post exertional malaise, excessive thirst, and gastrointestinal symptoms.

#### 2.3.2. Inflammatory Markers and Oxidized LDL

Fasting blood (>8 h fast) was obtained and sent to a CLIA-certified laboratory at UHCMC for the measurement of lipids, creatine kinase, glucose and insulin with Homeostatic Model Assessment for Insulin Resistance (HOMA-IR) derived from the latter 2 tests. Additional blood was processed within 2 h, aliquoted, and stored at −80 °C until shipment on dry ice to Dr. Funderburg’s laboratory, Ohio State University, for the measurement of biomarkers using the enzyme-linked immunosorbent assay (ELISA). The following were measured: markers of monocyte activation soluble CD14 and CD163 (sCD14 and sCD163), markers of inflammation high-sensitivity C-reactive protein (hs-CRP), interleukin (IL)-6, interferon-gamma inducible protein of 10 kDa (IP-10), tumor necrosis factor receptor 1 and 2 (TNF-RI and TNF-RII), and vascular cell adhesion molecule (VCAM) using R&D Systems (Minneapolis, MN, USA); oxidized low-density lipoprotein (oxLDL) using kits (Upsala, Mercodia, Sweden); and D-dimer (Diagnostica Stago, Parsippany, NJ, USA).

#### 2.3.3. Body Composition Measurements

Both DXA#1 and DXA#2 were performed on the same machine for all participants, using the Hologic, Horizon A, 5.6.0.4 DXA system. DXA quantified the total fat, limb fat, and trunk fat, total LBM, and total BMD.

### 2.4. Statistical Analysis

The characteristics of study participants were described using the frequency (*n*) and percentage (%) for categorical variables and the mean ± standard deviation (std) or median and interquartile range (IQR) for continuous variables. Differences between groups at DXA#1 were assessed using independent *t*-tests, Kruskal–Wallis tests, or chi-square tests. To account for the difference in time between DXA#1 and DXA#2, longitudinal generalized linear mixed models with random intercepts were used to estimate within- and between-group changes in inflammation. DXA outcomes were transformed to reflect annualized change and annualized percentage (%) change. The annualized change was calculated by dividing the mean difference in each outcome between DXA measures by the difference in follow-up time $\left(\frac{DXA\;2-DXA\;1}{{Visit\;Date}_{DXA\;2}-V{isit\;Date}_{DXA\;1}}\right)$. The annualized % change was derived by dividing the annualized change by the mean outcome at DXA 1 $\left(\frac{annualized\;Outcome}{DXA\;1\;Outcome}*100\right).$ The adjusted models included COVID-19 and HIV status, age at DXA#1, sex, race, and the difference between DXA#1 and DXA#2, and each inflammation marker was modeled separately. All analyses were conducted using SAS 9.4 (SAS Inc., Cary, NC, USA), and *p*-values less than alpha < 0.05 were considered statistically significant.

## 3. Results 

### 3.1. Characteristics of Study Participants

A total of 160 prospectively enrolled adults were included in the analysis. The average age was 43.2 ± 14.5 years, 32.5% were female (*n* = 52), and 51.8% (*n* = 83) were of a non-white race. Eighty participants (50%) had a documented COVID-19 infection (COVID-19+) between DXA#1 and DXA#2 (Table 1). All participants living with HIV (HIV-positive) were on antiretroviral therapy with a viral load below the limit of detection. At the pre-pandemic visit, the COVID-19+ group had a smaller proportion of smokers [53% (COVID-19-) vs. 30% (COVID-19+); *p* = 0.002] and HIV-positive participants [78% (COVID-19-) vs. 58% (COVID-19+); *p* = 0.004s]. However, there was not enough evidence to suggest any differences (*p* > 0.05) in weight, HOMA-IR, alcohol consumption, weekly physical activity, HDL, non-HDL, creatinine kinase, or body composition measures between the groups. The median number of days between DXA measurements among the COVID-19-group was 500 days, and it was 899 days among the COVID-19+ group (*p* < 0.0001). Among the COVID-19-infected group, 55% (*n* = 44) had PASC (COVID-19+ PASC+). Among the COVID-19+ group without PASC (COVID-19+ No PASC), the median number of days between DXA visits was 743 (IQR: 420, 1560), and it was 915.5 (IQR: 673.5, 1489.5) days among the COVID-19+ PASC+ subjects. The five most frequently reported symptoms were fatigue (58.2%), trouble thinking or concentrating (51.2%), shortness of breath (44.2%), feeling sad, down, depressed, or anxious (39.5%), and feeling pain in any body part (34.9%). During the acute COVID-19 illness, 92% of the COVID-19+ group had managed their symptoms at home without hospitalization; only one participant was hospitalized for 2 days without ICU admission. Additionally, only 3.75% received steroids during the study period, none of them specifically for COVID-19.

### 3.2. Changes in Inflammation

Accounting for the differences in time between DXA#1 and DXA#2 (Table 2), an increase in VCAM (71.5 ng/mL; *p* = 0.06) and oxLDL (33.5 U/L; *p* < 0.001) among the COVID-19+ group were observed. Within the COVID-19-group, a decrease in TNF-RII (−877.9 pg/mL; *p* < 0.001) and D-dimer (−473.1 ng/mL; *p* = 0.04) were observed. Comparing the changes in inflammation over time between the 2 groups, there is evidence that COVID-19 infection has an effect on VCAM (*p* = 0.02), TNF-RII (*p* < 0.0001), and oxLDL (*p* < 0.0001). 

### 3.3. Annualized Change in Body Composition

In Figure 1, COVID-19-participants had larger annualized increases in trunk fat (g) [922.5 vs. 159.7 (Δ = −762.9); *p* = 0.01], total fat (g) [1564.3 vs. 199.9 (Δ = −1364.3); *p* = 0.2], and LBM (g) [974.9 vs. −64.5 (Δ = −1039.4); *p* = 0.0002], compared to the COVID-19+ group. There was no effect of HOMA-IR or alcohol consumption on measures of body composition. Because of the potential effect of HIV status on body composition, we adjusted for both HIV and COVID-19 status. In the adjusted models (Table 3), the between-group differences in trunk fat (*p* = 0.004) and LBM (*p* = 0.001) remained. Among inflammation markers, only hsCRP was associated with trunk fat (*p* = 0.01) and total fat mass (*p* = 0.04). 

### 3.4. Effect of PASC Status on Body Composition among COVID-19 Survivors

Among COVID-19+ subjects without PASC, the median annualized trunk fat was 150.2 (IQR: −217.6, 743.1), the total fat mass was 166.2 (IQR: −623.6, 1258.7), and the LBM was 113.3 (IQR: −371.9, 965.2). Among COVID-19 survivors with PASC+, the median annualized trunk fat was 300 (IQR: −449.9, 1159.3), the fat mass was 471 (IQR: −1073.4, 1764.8), and the LBM was −242.4 (IQR: −1203.6, 483.2). However, the differences in annualized trunk fat, total fat mass, or LBM were not significantly different (*p* > 0.05 for all) between those with and without PASC.

## 4. Discussion 

In this prospective longitudinal study, we investigated the long-term changes in the body composition of COVID-19 survivors compared to a contemporary control group. We were the first to dissociate the effect of COVID-19 infection from the indirect effect of the pandemic on body composition. Each patient underwent a pre-pandemic whole body DXA scan and a post-pandemic DXA scan within five years. After adjusting for possible confounders, we found that both groups had increased their total fat, trunk fat, and weight during the study period. The changes were even more prominent in the COVID-19-negative group. These findings are in line with the longitudinal study by Marcos-Pardo which showed a significant increase in trunk fat mass and total fat mass after a 13 week-lockdown in Spanish older women [6]. A meta-analysis published in 2021 also demonstrated an increase in weight during the pandemic [5]. Furthermore, in a meta-analysis, some athletes were found to gain weight during the COVID-19 pandemic, despite all of their efforts to maintain their physical performance [23].

These results can be explained by the impact of the pandemic, regardless of COVID-19 infection status, while also taking into account the decline in health caused by a sedentary lifestyle, as well as the lockdown measures [24]. Although our study did not specifically consider the participants’ lifestyle habits during the pandemic, previous research has shown a significant reduction in physical activity, along with an increase in snacking frequency and carbohydrate and fast-food consumption during this period [24,25,26]. 

On the other hand, our study found that individuals who had contracted COVID-19 during the study period showed a slight decrease in LBM over time, while those who remained uninfected experienced an increase in LBM. Our results align with a cross-sectional study conducted by Ramirez-Velez et al. which found that individuals diagnosed with long-COVID syndrome had significantly lower levels of total and appendicular lean mass tissue and inferior muscle strength parameters compared to controls [8]. In addition, our study’s findings support a previous cross-sectional study by Lopez-Sampalo which revealed higher percentages of sarcopenia among older individuals three months after COVID-19 infection with reduced muscle strength. It is worth noting that muscle strength in the mentioned study was measured via dynamometry using grip strength, while body composition was obtained using bioelectrical impedance [7].

The observed difference in LBM between the two groups in our study may be due to the direct effect of SARS-CoV-2 infection on muscle cells given that only the COVID-19+ group showed a deterioration in LBM compared to uninfected individuals. The majority of our participants were not taking steroids during the study period, suggesting that our findings are not confounded and are independent of steroid use. Studies have shown that SARS-CoV-2 enters muscle cells through ACE receptors, disrupting their homeostasis and causing inflammation within these cells, which affects their function during acute infection, causing a wide range of muscle symptoms [14,17,18]. Although the etiology of PASC remains unclear, a recent study used electromyography and muscle biopsies to investigate the cause of post-COVID-19 fatigue, myalgia, and weakness that lasted up to 14 months after infection. This study found muscle fiber atrophy, regeneration, mitochondrial alterations, inflammation, and capillary damage, changes that suggest that SARS-CoV-2 may target the skeletal muscles, which may lead to long-term myopathy, even in cases of mild or moderate acute infection [27]. Another recent study supports this evidence by showing pathophysiological changes in muscle biopsies taken from individuals diagnosed as having PASC with post-exertional malaise. The biopsies showed a reduction in skeletal muscle mitochondrial enzyme activity together with a blunted T-cell response following acute exercise, an increased accumulation of amyloid-containing deposits, and signs of severe muscle tissue damage [28].

Our study did not find significant differences in annualized LBM, trunk fat, and total fat between individuals who had PASC and those who seemingly underwent a full recovery after COVID-19 infection. This suggests that individuals who have had COVID-19 may experience the same changes in body composition, including muscle involvement, as individuals who did develop PASC, despite the absence of symptoms. Further studies investigating muscle involvement in PASC should always have a control group of COVID-19 survivors who did not develop PASC. 

Finally, our study did not demonstrate any change in total BMD between groups, although total BMD is a less sensitive marker of worsening in BMD when compared to lumbar or hip BMD. Previously, Berktas reported a decrease in BMD after COVID-19 infection [29]. Similarly, Elmedany observed a significant decline in BMD among individuals with osteoporosis 9 months after contracting COVID-19 [30]. These results can be explained by the fact that inflammatory mediators affect the expression of RANK (osteoclast receptor) and RANKL (osteoblast ligand), thereby disrupting bone remodeling and increasing bone resorption [31,32]. Further studies are necessary to evaluate the longer-term effects of COVID-19 on bone health using hip and lumbar BMD.

It is known that COVID-19 survivors have elevated levels of several inflammation markers [33]. We measured different biomarkers of inflammation to study their relationship with changes in body composition. We did show that COVID-19 survivors experienced increases in several markers of inflammation when compared to the COVID-19-negative group, specifically sTNF-RII, VCAM and oxLDL; however, none of these markers were associated with body composition measurements. Only hs-CRP was associated with changes in total and trunk fat, a finding known to occur in the general population outside of COVID-19 infection. It is notable that our finding of increased oxLDL in COVID-19 survivors aligns with previous research [34,35], highlighting a correlation between this marker and COVID-19 infection, in particular in individuals who develop PASC.

Understanding the relationship between COVID-19 infection and alterations in body composition can help healthcare professionals in implementing early strategies to help prevent these long-term changes as well as developing personalized exercise and nutrition plans to optimize body composition and reduce the risk of metabolic complications associated with COVID-19 recovery.

There are a few limitations to our study that should be addressed. First, we did not report on lifestyle habits such as detailed physical activity and diet during the study period. Second, our study estimated total body BMD, which is not sensitive in detecting small changes in BMD; future studies should use lumbar spine and hip BMD. Additionally, as with any cohort study, in spite of our efforts to minimize selection bias and residual confounding, we acknowledge that our findings should be interpreted with this in mind. Finally, the study consisted of a higher proportion of male participants compared to females Therefore, it would be inappropriate to generalize these results to the entire population.

## 5. Conclusions

In conclusion, we demonstrated changes in body composition in a group of COVID-19 survivors when compared to a group of individuals who did not have a COVID-19 infection during the study period. Both groups exhibited increases in total fat, trunk fat, and weight over the course of the study, potentially linked to lifestyle changes during the pandemic. Notably, all COVID-19 survivors regardless of PASC status witnessed a decline in their lean body mass and an increase in several inflammatory biomarkers over the same period, suggesting a direct impact of COVID-19 on muscle, possibly related to enhanced inflammation. Further larger and longer-term studies should assess the effect on body composition of COVID-19 infection and the pathological mechanisms associated with these changes. 

## Figures and Tables

**Figure 1 nutrients-16-01364-f001:**
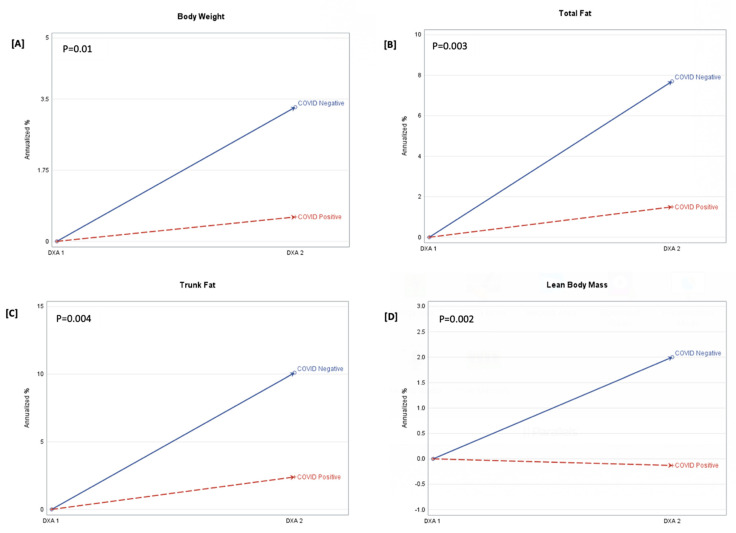
Annualized percentage (%) change in body weight, total fat, trunk fat, and lean body mass. FOOTNOTE: Annualized percentage (%) change (APC) = (annualized change/DXA1)*100. *p*-value is the between-group probability. The derived APC is shown at DXA 1(baseline) and DXA 2, stratified by the COVID-19-negative and COVID-19-positive groups. Panel (**A**) is the APC in body weight (kg), panel (**B**) is the APC in total fat (g), panel (**C**) is the APC in trunk fat (g), and panel (**D**) is the APC in lean body mass (g).

**Table 1 nutrients-16-01364-t001:** Baseline characteristics (at DXA#1) among COVID-19 survivors and COVID-19-negative participants.

	COVID-19− (*n* = 80)	COVID-19+ (*n* = 80)	*p*-Value
	*n* (%), Median (IQR), or Mean ± std		
Characteristics			
Age (years)	43.2 ± 15.4	42.9 ± 13.6	0.9
Female Sex	25 (28.4)	27 (33.8)	0.5
Non-white Race *	55 (63)	33 (41.3)	0.01
Weight (kg)	83 ± 20.3	87 ± 22.2	0.4
Body Mass Index (kg/m^2^)	28.4 ± 6.3	29.7 ± 6.8	0.2
Current Smoker (Yes)	47 (53.4)	24 (30)	0.002
Current Alcohol Use (Yes)	57 (71.3)	56 (70)	0.9
Physical Activity (min/week)	2520 (870, 3735)	2310 (1260, 3525)	0.7
Mid-Waist Circumference (cm)	95.8 ± 17.6	98.2 ± 16.9	0.3
HIV Status (+)	69 (78)	46 (58)	0.004
Days between DXA 1 and 2	500 (399, 936)	899 (569, 1495)	0.0001
HDL (mg/dL)	46.2 ± 12.9	48.1 ± 12.9	0.4
non-HDL	120.5 ± 34.5	129.2 ± 37.5	0.07
Creatine Kinase (uL)	207.3 ± 159.5	167.9 ± 117.2	0.2
Homa-IR	2.6 ± 2.2	3.7 ± 4.9	0.6
DXA Markers			
Total Body BMD	1.2 (1.1, 1.2)	1.1 (1.1, 1.2)	0.33
Trunk Fat (g)	11,806.4 (6907.5, 17,260.8)	13,206.3 (10,113.7, 17,745.4)	0.07
Total Fat (g)	24,921.4 (16,809.8, 34,842.2)	26,782.6 (21,737.7, 34,204.1)	0.08
Total LBM (g)	52,423 (45,930.1, 58,476.8)	52,956.2 (45,096.4, 62,194.1)	0.4
Inflammation Markers			
IL-6 (pg/mL)	2.8 (1.8, 5.6)	2.5 (1.5, 3.8)	0.1
VCAM (ng/mL)	830.2 (673.7, 1018.5)	739.8 (642.6, 963.1)	0.06
TNF-RI (pg/mL)	1046.8 (847.8, 1328.6)	1006.4 (771.7, 1141.1)	0.05
TNF-RII (pg/mL)	2961.1 (2364, 4366.7)	2190.2 (1874, 2898.2)	<0.0001
hsCRP (ng/mL)	3512.4 (1287.5, 7684.6)	2696.8 (880.4, 8651.2)	0.3
IP10 (pg/mL)	181.4 (135.4, 328.5)	124.5 (97.5, 196.7)	0.001
D-dimer (ng/mL)	411.6 (274.8, 600.2)	316.4 (195.8, 517.1)	0.02
oxLDL (U/L)	43.4 (35.3, 52.9)	50.9 (38.1, 65.7)	0.02
sCD14 (ng/mL)	1711.9 (1435.1, 2117.8)	1592.5 (1305.9, 1949.8)	0.1
sCD163 (ng/mL)	708.6 (530.9, 1148.4)	624.9 (410.2, 851.2)	0.01

* Includes African American, Asian, Hispanic, and Other; Abbreviations: HDL = high-density liproprotein; IL-6 = interleukin-6; VCAM = vascular cell adhesion molecule-1; TNF-RI = tumor necrosis factor receptor-1; TNF-RII = tumor necrosis factor receptor-2; hsCRP = high-sensitivity C-reactive protein; I-CAM = intercellular adhesion molecule-1; IP10 = interferon-gamma inducible protein of 10 kDa; oxLDL = oxidized LDL; sCD14 = soluble CD14; sCD163 = soluble CD163.

**Table 2 nutrients-16-01364-t002:** Changes in inflammation between DXA 1 and DXA 2 by COVID-19 status.

	COVID-19−	COVID-19+	*p*-Value (between Group)
	Mean ± Std (*p*-Vaue) *		
VCAM	−53.03 ± 37.3 (0.2)	71.5 ± 38.4 (0.06)	0.02
TNF_RI	−53.2 ± 46.0 (0.3)	62.5 ± 47.0 (0.2)	0.08
TNF-RII	−877.9 ± 181.9 (<0.001)	41.0 ± 185.9 (0.8)	<0.0001
hsCRP	−2390.3 ± 1668.5 (0.2)	−312.2 ± 1693.0 (0.8)	0.4
D-dimer	−473.1 ± 230.1 (0.04)	93.2 ± 235.1 (0.7)	0.08
oxLDL	47.2 ± 42.4 (0.3)	33.5 ± 45.1 (<0.001)	<0.0001

* *p*-value for within-group comparisons. Abbreviations: VCAM = vascular cell adhesion molecule-1; TNF-RI = tumor necrosis factor receptor-1; TNF-RII = tumor necrosis factor receptor-2; hsCRP = high-sensitivity c-reactive protein; oxLDL = oxidized LDL.

**Table 3 nutrients-16-01364-t003:** Annualized change in trunk fat, total fat, and lean body mass.

	Trunk Fat				Total Fat Mass				Total LBM			
	Unadjusted		Adjusted		Unadjusted		Adjusted		Unadjusted		Adjusted	
COVID-19 Status (+ vs. −)	−762.9 ± 279.02	(0.01)	−895.8 ± 301.6	(0.004)	−1364.3 ± 484.2	(0.2)	−1607.6 ± 523.7	(0.2)	−1039.4 ± 269.7	(0.0002)	−963.3 ± 293.0	(0.001)
VCAM	−1.02 ± 0.42	(0.02)	−1.1 ± 0.42	(0.01)	−1.97 ± 0.72	(0.2)	−2.11 ± 0.72	(0.2)	−0.7 ± 0.41	(0.1)	−0.7 ± 0.41	(0.1)
TNF-RI	−0.06 ± 0.4	(0.9)	0.17 ± 0.4	(0.7)	−0.4 ± 0.7	(0.6)	−0.05 ± 0.7	(0.9)	−0.6 ± 0.4	(0.1)	−0.2 ± 0.4	(0.6)
TNF-RII	−0.14 ± 0.1	(0.1)	−0.06 ± 0.1	(0.5)	−0.3 ± 0.2	(0.4)	−0.13 ± 0.2	(0.6)	−0.13 ± 0.1	(0.1)	−0.02 ± 0.1	(0.8)
hsCRP	0.03 ± 0.01	(0.01)	0.03 ± 0.01	(0.01)	0.03 ± 0.02	(0.3)	0.04 ± 0.02	(0.04)	0.01 ± 0.01	(0.2)	0.02 ± 0.01	(0.1)
D-dimer	−0.003 ± 0.07	(0.9)	−0.01 ± 0.1	(0.8)	−0.03 ± 0.1	(0.8)	−0.04 ± 0.1	(0.8)	−0.04 ± 0.1	(0.6)	−0.04 ± 0.1	(0.6)
oxLDL	0.004 ± 0.01	(0.4)	0.01 ± 0.01	(0.1)	0.01 ± 0.01	(0.5)	0.02 ± 0.01	(0.3)	−0.001 ± 0.01	(0.8)	0.003 ± 0.01	(0.5)

Adjusted models included COVID-19 and HIV status, age at DXA 1, sex, race, and the difference between pre-pandemic and post-pandemic markers of inflammation; Abbreviations: VCAM = vascular cell adhesion molecule-1; TNF-RI = tumor necrosis factor receptor-1; TNF-RII = tumor necrosis factor receptor-2; hsCRP = high-sensitivity C-reactive protein; oxLDL = oxidized LDL.

## Data Availability

Restrictions apply to the datasets. The datasets presented in this article are not readily available because the data are part of an ongoing study. Requests to access the datasets should be directed to the corresponding author by email.

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
