# Peer review of "The Long-Term Effect of COVID-19 Infection on Body Composition"

_nutrients, 2024, doi:10.3390/nu16091364_

Round 1
Reviewer 1 Report
Comments and Suggestions for Authors
The manuscript presents an insightful investigation into the long-term effects of COVID-19 on body composition compared to a control group, with a focus on pre-pandemic and post-pandemic DXA scan data. Below are some comments and suggestions for improving the manuscript:
-
Introduction:
- - Clear rationale
-
Methods:
- -No COVID-19 severity assessment, no power analysis,
- - No data about diet
- Please specify why these covariates were included in the adjusted models.
-
Results:
- Well organized
-
Discussion:
- Good integration of previous studies
- Authors highlight the novelty of the this study.
- Please Address potential limitations of the study, such as selection bias, loss to follow-up, and the possibility of residual confounding.
Author Response
We would like to thank the reviewers for their thorough assessment of our manuscript. We have considered their feedback very carefully, and have made the necessary adjustments during the revision process. We are pleased to present an improved version of our manuscript.
Reviewer#1
We express our gratitude to Reviewer #1 for their positive feedback. We have carefully noted the remaining comments and they are addressed below:
- Comment 1: “No COVID-19 severity assessment.”
RESPONSE to comment 1:
As already mentioned in the results section, 92% of study participants managed their symptoms at home. Only one participant was hospitalized for 2 days without ICU admission and 3.75% received steroids during the study period, none of them specifically for COVID-19. This underscores the predominance of minor symptoms among our participants. Additionally, in the baseline characteristics of the Methods section, it was specified that a comprehensive medical history was obtained from the medical records of all participants, including details pertinent to COVID-19 infection and course of illness.
- Comment 2: “No Power Analysis.”
RESPONSE to comment 2:
No power calculation was needed given the design of the study.
We have updated the study design paragraph in the manuscript (highlighted in yellow) to clarify the inclusion/exclusion criteria and power calculation as follows:
“Participants with both available pre-pandemic data, including whole body DXA scan, obtained between 2017 and 2019, and post-pandemic DXA data were selected for this study. 160 participants met these criteria consisting of 80 COVID-19 survivors and 80 participants who never had SARS-CoV-2 infection. The whole-body DXA scan available from the pre-pandemic period is referred to as DXA#1 while the post-pandemic DXA as DXA#2. No power calculation was needed given the design of the study. Comprehensive demographics, clinical data including medical history, and inflammation markers were obtained at DXA#1 and DXA#2. Fasting lipids and creatine kinase (CK) levels were assessed in real-time in a CLIA-certified laboratory at University Hospitals Cleveland Medical Center (UHCMC), Cleveland, Ohio.”
- Comment 3: “No data about diet.”
RESPONSE to comment 3:
Agree. This was already acknowledged as a limitation in the discussion section.
- Comment 4: “Please specify why these covariates were included in the adjusted models.”
RESPONSE to comment 4:
We a priori decided to include in adjusted models the covariates that are known to be associated with changes in body composition.
- Comment 5: “Please address potential limitations of the study, such as selection bias, loss to follow-up, and the possibility of residual confounding.”
RESPONSE to comment 5:
Additional limitations were added in the discussion section.
Please refer to the discussion section of the manuscript for the updated paragraph (highlighted in yellow).
The added paragraph is:
“Additionally, as with any cohort study, in spite of our efforts to minimize selection bias and residual confounding, we acknowledge that our findings should be interpreted with this in mind.”
Reviewer 2 Report
Comments and Suggestions for Authors
This is a well-written and interesting paper assessing long-term changes in body composition in COVID–19 survivors compared to a contemporary control group. The content is very original and well-argued. Body composition was assessed through the whole-body DXA scan and the whole sample consists of 180 participants. Despite the study's interest, there are some weaknesses.
From a methodological point of view, I identified the following three weaker points:
1- The reporting of anthropometric values with mixed sexes does not seem correct, all the more so given the difference in numerosity. From an anthropometric point of view, males and females differ significantly in weight and body composition parameters. Regarding the parameters obtained by DXA in particular, I suggest you consult the literature. See, for example, https://doi.org/10.1038/s41430-020-0596-5. Therefore, the tables need to be revised by reporting these characters separately by sex.
2- Regarding measurements of stature and weight, what is given on line 116 does not seem sufficient (“Weight and height were measured using standardized protocol”). If you do not want to dwell on the measurement methods and instruments used, you need to at least cite some traditional anthropometry manuals (e.g., Lohman, T.G.; Roche, A.F.; Martorell, R. Anthropometric standardization reference manual. Human Kinetics Books: Champaign, 1988).
3- The percentage of women examined corresponds to less than one-third of the total sample. This represents a clear weakness of the study that at the very least should be reported among the limitations.
Author Response
We would like to thank the reviewers for their thorough assessment of our manuscript. We have considered their feedback very carefully, and have made the necessary adjustments during the revision process. We are pleased to present an improved version of our manuscript.
Reviewer#2
We express our gratitude to Reviewer #2 for their positive feedback. We have carefully noted the remaining comments and they are addressed below:
- Comment 1: “The reporting of anthropometric values with mixed sexes does not seem correct, all the more so given the difference in numerosity. From an anthropometric point of view, males and females differ significantly in weight and body composition parameters. Regarding the parameters obtained by DXA in particular, I suggest you consult the literature. See, for example, https://doi.org/10.1038/s41430-020-0596-5. Therefore, the tables need to be revised by reporting these characters separately by sex.”
RESPONSE to Comment 1:
We agree and acknowledge Reviewer #2's observation regarding the differences in weight and body composition parameters between males and females. However, our interest was not looking at sex differences, and given there was no difference in the proportion of male and female sex between groups as displayed in Table 1, sex, more importantly, did not affect our outcomes.
If despite this argument the reviewer still thinks that the tables need to be revised by reporting these characters separately by sex, we could provide a new table showing these parameters separately by sex and add it as supplementary material.
- Comment 2: “Regarding measurements of stature and weight, what is given on line 116 does not seem sufficient (“Weight and height were measured using standardized protocol”). If you do not want to dwell on the measurement methods and instruments used, you need to at least cite some traditional anthropometry manuals (e.g., Lohman, T.G.; Roche, A.F.; Martorell, R. Anthropometric standardization reference manual. Human Kinetics Books: Champaign, 1988).”
RESPONSE to comment 2: we appreciate your comment and great suggestion regarding measurements of stature and weight. We have indeed used LOhman’s methods and updated the manuscript with a citation of Lohman, T.G.; Roche, A.F.; Martorell, R. Anthropometric standardization reference manual. Human Kinetics Books: Champaign, 1988.
Please refer to the manuscript for added citation on line 127 (highlighted in yellow and line 390 in the references section)
- Comment 3: “The percentage of women examined corresponds to less than one-third of the total sample. This represents a clear weakness of the study that at the very least should be reported among the limitations.”
RESPONSE to comment 3:
In our study, we aimed to recruit an equal number of male and female participants. However, we encountered difficulties in recruiting female participants specifically, resulting in only 32% female participation. We acknowledge this as a limitation of our study, and it should be noted that the results may not be generalizable to the entire population.
In the revised manuscript in the discussion section (highlighted in yellow), an additional sentence has been included regarding this point as follows:
“The study consisted of a higher proportion of male participants compared to females Therefore, it would be inappropriate to generalize these results to the entire population”
Reviewer 3 Report
Comments and Suggestions for Authors
The manuscript provides significant insights into how COVID-19 infection affects body composition over the long term. However, to increase the likelihood of acceptance, the Authors of the study should clarify how the sample was selected and provide information on their demographics. They should also control for any confounding factors, ensure that the timing of DXA scans is clearly stated, conduct thorough and robust statistical analysis, address generalizability, and suggest recommendations for future research. Lastly, the manuscript should emphasize the clinical implications of the findings.
Author Response
We would like to thank the reviewers for their thorough assessment of our manuscript. We have considered their feedback very carefully, and have made the necessary adjustments during the revision process. We are pleased to present an improved version of our manuscript.
Reviewer#3
We thank Reviewer #3 for their positive feedback. We have carefully noted the remaining comments and they are addressed below:
- Comment 1: “To increase the likelihood of acceptance, the authors of the study should clarify how the sample was selected and provide information on their demographics.”
RESPONSE to comment 1:
The study design paragraph has been revised to clarify inclusion criteria and a thorough description of participant recruitment is already outlined in section 2.2-2.3. Extensive information on participant demographics/characteristics is outlined in Table 1 and the opening paragraphs in the results section.
The revised Study Design paragraph is as follows (highlighted in yellow in the manuscript):
“Participants with both available pre-pandemic data, including whole body DXA scan, obtained between 2017 and 2019, and post-pandemic DXA data were selected for this study. 160 participants met these criteria consisting of 80 COVID-19 survivors and 80 participants who never had SARS-CoV-2 infection. The whole-body DXA scan available from pre-pandemic period is referred to as DXA#1 while the post-pandemic DXA as DXA#2. No power calculation was needed given the design of the study. Comprehensive demographics, clinical data including medical history, and inflammation markers were obtained at DXA#1 and DXA#2. Fasting lipids and creatine kinase (CK) levels were assessed in real time in a CLIA-certified laboratory at University Hospitals Cleveland Medical Center (UHCMC), Cleveland, Ohio.”
- Comment 2:
They should also control for any confounding factors, ensure that the timing of DXA scans is clearly stated, conduct thorough and robust statistical analysis, address generalizability, and suggest recommendations for future research. Lastly, the manuscript should emphasize the clinical implications of the findings.
RESPONSE to comment 2:
In our study, we controlled for numerous confounding factors, including COVID and HIV status, age at DXA #1, sex, race, and variations between pre-pandemic and post-pandemic markers of inflammation. As outlined in the methods section, DXA #1 was conducted before the pandemic, from 2017 to 2019, with variations in individual time points. DXA #2, on the other hand, was performed at least one-year post-COVID infection for the COVID+ group. The mean duration between DXA #1 and #2 was also detailed in the results section. Due to the difference in timing of DXA scans, we conducted annualized changes in different body composition parameters for accurate comparisons.
In our manuscript, we suggested for future research to conduct larger and longer studies to expand the scope of participant recruitment and extend the duration of follow-up to allow for a comprehensive evaluation of the trajectory of body composition changes over time post-COVID infection. Additionally, we suggested conducting histopathological studies to correlate clinical findings with pathological changes observed in tissues that could provide insights into the underlying mechanisms driving alterations in body composition.
Lastly, in the revised manuscript, we have incorporated a small paragraph to emphasize the clinical implications of our findings. The updated paragraph is as follows (highlighted in yellow in the discussion section of the manuscript):
“Understanding the relationship between COVID-19 infection and alterations in body composition can help healthcare professionals in implementing early strategies to help prevent these long-term changes as well as developing personalized exercise and nutrition plans to optimize body composition and reduce the risk of metabolic complications associated with COVID-19 recovery.”
Round 2
Reviewer 2 Report
Comments and Suggestions for Authors
I am satisfied with the answers provided by the authors. The manuscript is ready for publication.